# ENHANCING LLM REASONING WITH RETRIEVAL-AUGMENTED LOGICAL CHAINS AND TEST-TIME ADAPTATION

## ABSTRACT

Large language models (LLMs) excel on knowledge-intensive tasks but often fail on complex, multi-step reasoning requiring explicit inference and logical coherence. Retrieval-augmented generation (RAG) grounds outputs in external text, yet retrieved content is typically unstructured and misaligned with step-wise reasoning. We introduce LogicalChain, a framework that explicitly integrates structured logical chains—interpretable, step-by-step derivations linking context to conclusions. We build a large corpus of chains from domain-rich sources (e.g., expert guidelines, worked solutions) and train a contrastive retriever to fetch task-relevant inference paths. To close the instance–step misalignment at inference, we propose *TTT–RAG*, a test-time adaptation pipeline that fine-tunes the LLM on retrieved chains and documents *during* inference, tailoring behavior without updating global weights. Experiments show consistent gains across **medical** and **general multi-hop** domains: on MedQA, TTT–RAG lifts Qwen2.5–7B–Instruct from 53.8% to 70.1% (14B: 73.8%), and on MedMCQA to 62.1% (14B: 64.3%). Beyond the medical domain, TTT–RAG improves general multi-hop reasoning, reaching 45.1/42.8 (7B) and 48.5/44.6 (14B) on MultiHopQA/2Wiki, surpassing strong CoT baselines (e.g., rStar) and RAG systems (MedRAG, i-MedRAG). These results indicate that injecting structured reasoning pathways at test time yields scalable, interpretable, and state-of-the-art performance for complex reasoning tasks across domains [1].

## 1 INTRODUCTION

Solving problems through structured reasoning is a hallmark of human intelligence McCarthy (1959); Weizenbaum (1966); Winograd (1972); Shortliffe (2012); Lenat and Guha (1989). Whether tackling complex math, diagnosing patients, or weighing hypotheses, humans construct interpretable chains of inference that link observations to conclusions. LLMs can mimic this behavior on many benchmarks, yet they degrade on domain-specific tasks that demand expert, multi-step reasoning Hodel and West (2023); Dasgupta et al. (2022); Zhang et al. (2024) (Figure 1), especially when the problem is novel or compositional Liu et al. (2023); Dziri et al. (2023). A central cause is that pretraining data overrepresents surface heuristics and underrepresents formal reasoning procedures Sunstein and Hastie (2015); Kahneman (2011). As a result, models default to shallow pattern matching and lack a mechanism to diagnose which intermediate assumptions are missing or wrong and to repair them by turning provisional chains into targeted evidence queries Dziri et al. (2023); Morishita et al. (2023); Guiaşu and Tindale (2018); Cheng et al. (2017). Without such diagnosis-and-repair, errors in intermediate steps persist unnoticed, producing hallucinations or logically incoherent answers on unfamiliar inputs.

Errors in complex reasoning are often step-local, so preventing propagation requires step-level verification at test time. Yet large models tend to leave intermediate premises implicit—a byproduct of pretraining objectives that reward next-token prediction and surface co-occurrence rather than explicit procedures—so the model cannot indicate which intermediate claim needs support. Hence any retrieval component intended to curb error propagation must be conditioned on explicit, step-indexed

---

[1] Our code and data: https://anonymous.4open.science/r/TTT-RAG-50F2/README.md

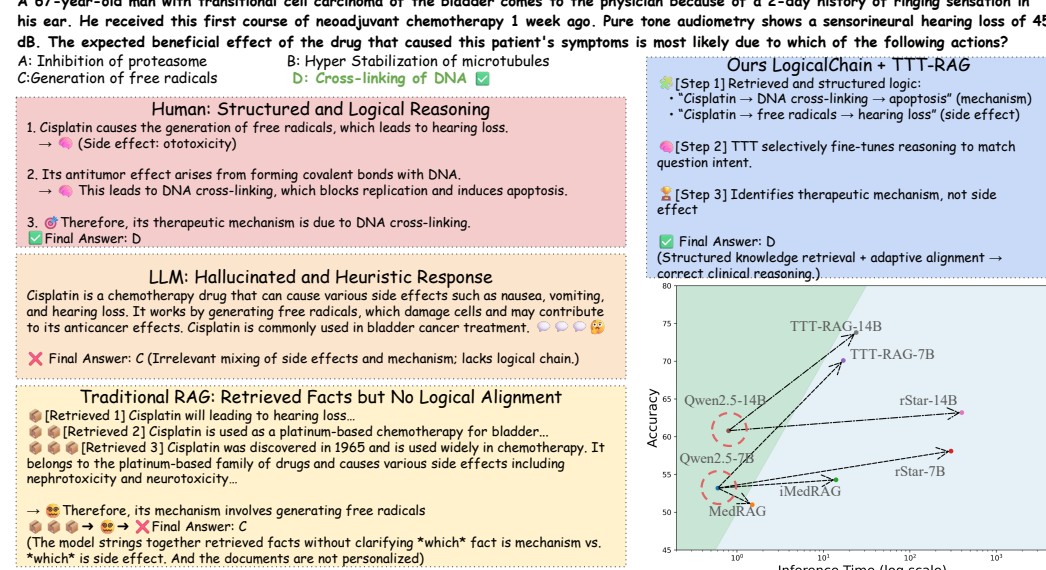

Figure 1: Motivation and Overview of LogicalChain + TTT-RAG Framework. (Left:) Solving clinical reasoning problems requires structured, interpretable inference—yet most LLMs fail to distinguish therapeutic mechanisms from side effects due to shallow heuristics and lack of logic chaining. Human physicians reason via modular logic (e.g., free radicals → hearing loss = side effect; DNA cross-linking → apoptosis = mechanism), enabling correct identification of causal chains. LLMs, in contrast, mix side effects with therapeutic logic, relying on memorized heuristics without structure. Traditional RAG retrieves scattered facts (e.g., chemotherapy side effects, DNA binding, etc.) but fails to organize them according to the reasoning goal (e.g., mechanism vs. symptom), highlighting the core research gap: retrieval without logical alignment and unpersonalized corpus often misleads LLMs. (Right:) We propose LogicalChain + TTT-RAG, a framework that enhances reasoning by: 1) Retrieving structured logical chains relevant to the question's intent to help the general domain model (in the green area) to answer other domain questions (in the blue domain) 2) Performing test-time training to align model generation with retrieved reasoning.

claims. Under this requirement, conventional RAG, which retrieves against the overall query, aligns evidence with the global question rather than the current step and often misses the weakest link (Lewis et al., 2020); interactive RAG can issue iterative requests (Xiong et al., 2024b), but when the model is overconfident it fails to generate the step-specific subquery needed to trigger retrieval, leaving the mistaken claim unchecked; and test-time scaling methods that compare or repair multiple chains (Qi et al., 2024) offer no mechanism—when failure stems from a missing fact—to insert the right premise at the right step. These limitations motivate two questions: **(RQ1)** how to elicit explicit, high-quality chains that surface step-level assumptions at inference time; and **(RQ2)** how to synchronize retrieval with these chains to verify each step, supply missing premises, and repair faulty links.

We propose **Test-Time-Training RAG (TTT-RAG)**, an end-to-end framework (Fig. 2) that operationalizes the two RQs by first eliciting and then verifying/repairing a chain. We introduce a provisional chain: a step-indexed, to-be-checked plan that makes the model's intended reasoning explicit before any evidence is injected. **For RQ1**, we build an off-line, retrieval-ready corpus of (Q, A, Chain) pairs from domain sources plus human-annotated chains, enforcing three principles: completeness of structured reasoning, entity-guided abstraction, and chain-enriched QA. A retriever/re-ranker is trained to surface step-indexed chain candidates. At inference, the same ranker assembles a small, question-conditioned minibatch of related (Q, A, Chain) exemplars (Compare-Ex. loop). We then perform light test-time training (TTT) on this minibatch to adapt the LLM so that it writes out its own provisional chain for the current instance, making intermediate hypotheses explicit. **For RQ2**, we treat each step of the TTT-elicited provisional chain as its own query and run step-level RAG to surface targeted logical chain—i.e., the specific premises, counter-examples, or factual cues implicitly needed at that step. These step-aligned docs are then fed back to the LLM to enable self correction of the chain. The retriever is also adapted at test time so that the fetched evidence tracks the evolving chain rather than only the original question. In summary, TTT-RAG uses one unified

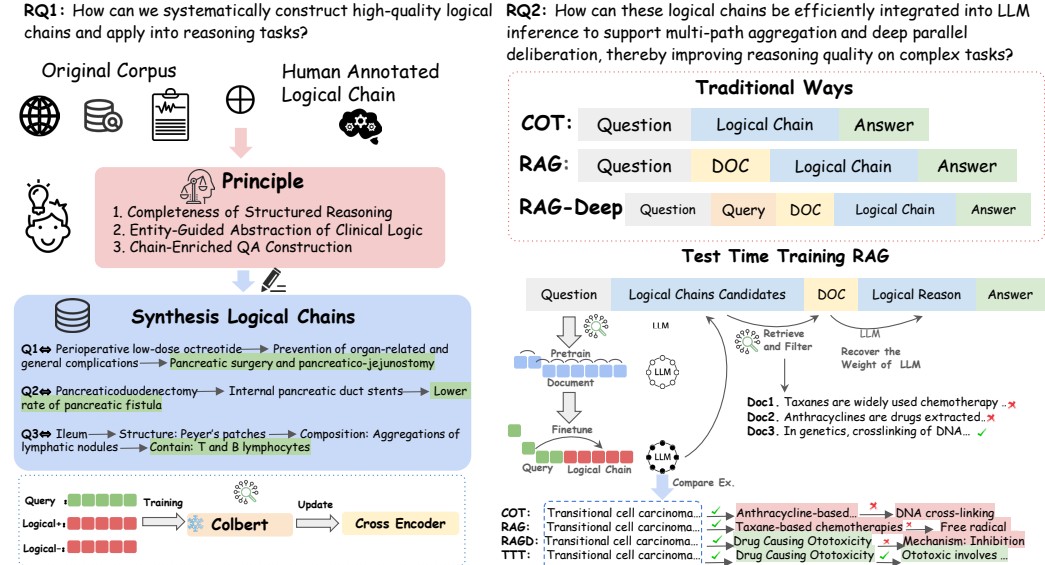

Figure 2: Overview of our proposed methodology, addressing two key research questions. (RQ1: Left) How can we systematically construct high-quality logical chains for reasoning tasks? We synthesize logical chains from both original corpora and human-annotated exemplars, guided by three core principles: (1) completeness of structured reasoning, (2) entity-grounded abstraction of domain logic, and (3) chain-enriched QA construction. These chains are derived from factual, domain-specific documents (e.g., medical guidelines or problem–solution sets), and are expressed as step-by-step derivations that map from question to answer. To support scalable integration, we train a dual-stage retriever: a ColBERT-based embedding retriever to shortlist logical chains, and a cross-encoder to refine selection. (RQ2: Right) How can logical chains be effectively integrated into LLM inference to improve reasoning? We compare three common paradigms—CoT, RAG, and RAG-Deep—with our proposed Test-Time Training RAG (TTT-RAG). Unlike prior methods that use unstructured documents or rigid decoding, TTT-RAG retrieves reasoning-aligned chains and performs test-time fine-tuning during inference to align generation with retrieved logic. The adapted model verifies chains through a document-grounded filtering stage and generates final answers with improved consistency and factual grounding. Case study comparisons (bottom row) show that TTT-RAG produces more accurate and interpretable reasoning compared to other methods.

(Q, A, Chain) corpus both to train the retriever and to assemble the test-time minibatch for TTT (addressing **RQ1**), and then applies step-indexed RAG with test-time retriever adaptation to check, complete, and—if necessary—revise the provisional chain (addressing **RQ2**).

We evaluate TTT-RAG on two representative domains that require logical reasoning: medical and mathematical reasoning. On medical benchmarks such as MedQA (Jin et al., 2021) and MedM-CQA (Pal et al., 2022), our method improves Qwen2.5-7B-Instruct (Yang et al., 2024) by over +16 points, outperforming strong baselines like MedRAG and rStar (Qi et al., 2024). On mathematical tasks like MATH Lightman et al. (2023) and Olympiad He et al. (2024) Bench, TTT-RAG also significantly surpasses comparably sized open-source models.

## 2 PROBLEM FORMULATION

**Hidden-chain view of reasoning.** Let $q \in \mathcal{Q}$ be a question and $M_\theta$ a (small) language model with parameters $\theta$. We posit that answering proceeds through an *unobserved* sequence of micro–inference states $\mathbf{z} = \langle z_1, \ldots, z_T \rangle \in \mathcal{Z}^T$ (the *latent logical chain*), followed by emission of an answer $y \in \mathcal{Y}$:

$$y = f_\theta(q, \mathbf{z}) \quad \text{with} \quad \mathbf{z} \sim p_\theta(\cdot \mid q).$$

Each adjacent pair $\tau_t = (z_t \to z_{t+1})$ is a *Wtransition* (claim, transformation, or sub-derivation). Errors typically arise at a local transition $\tau_{t^*}$ but remain hidden because $\mathbf{z}$ is not externalized.

**Why naive retrieval misses the failing step.** Standard retrieval conditions on the global query (either the input $q$ or the model's draft answer $y$), yielding document sets $\mathcal{D}^{(q)} = \text{TopK}\big(r_\phi(q)\big)$ and $\mathcal{D}^{(y)} = \text{TopK}\big(r_\phi(y)\big)$ from a corpus $\mathcal{C}$ using retriever $r_\phi : \mathcal{X} \to 2^{\mathcal{C}}$. Define a *step-alignment* functional $A : 2^{\mathcal{C}} \times \mathcal{T} \to [0, 1]$ that measures whether a set of documents directly relevant or irrelevant a specific transition $\tau$ (e.g., contains the premise, rule, entity relation, or counterevidence needed for that step). Empirically for the failing step $t^\star$,

$$A\big(\mathcal{D}^{(q)}, \tau_{t^\star}\big) \ll 1 \quad \text{and} \quad A\big(\mathcal{D}^{(y)}, \tau_{t^\star}\big) \ll 1,$$

because relevance to $q$ or $y$ is topical, not *step-indexed*; thus the wrong (or missing) micro-premise persists and the error propagates.

**Objective.** We seek a way to (i) elicit a provisional chain $\hat{\mathbf{z}} = \langle \hat{z}_1, \ldots, \hat{z}_T \rangle$ that reflects the model's intended intermediate states for this instance, and (ii) retrieve by transition, not by $q$ or $y$, so that evidence is locally aligned:

$$\forall t \in \{1, \ldots, T-1\} : \quad \mathcal{E}_t = \text{TopK}\big(r_\phi(q, \hat{\tau}_t)\big), \quad \hat{\tau}_t = (\hat{z}_t \to \hat{z}_{t+1}),$$

with the desideratum $A(\mathcal{E}_t, \tau_t) \gg A(\mathcal{D}^{(q)}, \tau_t), A(\mathcal{D}^{(y)}, \tau_t)$. Using step-aligned evidence $\mathcal{E}_{1:T-1}$, the model revises its reasoning to an updated chain $\mathbf{z}' = \langle z'_1, \ldots, z'_T \rangle$ and emits

$$y' = f_\theta\big(q, \mathbf{z}', \mathcal{E}_{1:T-1}\big),$$

aiming for improved correctness and faithfulness.

## 3 LOGICAL CHAIN GENERATION

To address the core limitation that SLMs seldom externalize the intermediate steps they implicitly rely on **(RQ1)**, we curate a training corpus that makes high-quality chains explicit and teaches the model what a faithful, step-indexed derivation looks like. Concretely, we construct logical chains that (i) decompose complex reasoning into atomic, verifiable transitions, (ii) stay semantically grounded in the entities actually present in the evidence, and (iii) are *used* to support question–answer pairs so that chains are not decorative but operational. These choices let us (a) train a chain retriever and chain-aware prompts/decoders that elicit a provisional chain at test time, and (b) provide step indices that later enable transition-conditioned retrieval.

### 3.1 DESIGN PRINCIPLES FOR CONSTRUCTING LOGICAL CHAINS

To effectively support structured reasoning, our dataset construction strategy is informed by the following design principles, grounded in formal reasoning completeness and clinical knowledge representation theory.

**Design Principle 1 (Structured Reasoning).** We represent a logical chain as a sequence of latent states $\mathbf{z} = \langle z_1 \to z_2 \to \cdots \to z_T \rangle$, where each $z_t$ is a micro–inference state and each *transition* $\tau_t = (z_t \to z_{t+1})$ captures one atomic step of reasoning. Let $\mathcal{A}$ denote the minimal set of atomic transition *types* (e.g., "premise→intermediate", "intermediate→conclusion"), and let $\mathcal{Z}$ be the set of valid chains. Any valid chain $\mathbf{z} \in \mathcal{Z}$ is decomposed into a finite sequence of atomic transitions $\mathbf{z} = \langle \tau_{i_1}, \tau_{i_2}, \ldots, \tau_{i_K} \rangle$ with $\tau_{i_k} \in \mathcal{A}$. This stepwise factorization (i) makes the chain interpretable and verifiable at the transition level and (ii) provides the indices $t$ we later use for transition-conditioned retrieval.

As a concrete example in the medical domain, an atomic step might be "symptom $\to$ diagnostic test (rule)" or "lab result $\to$ disease condition (conclusion)". While not all atomic steps immediately yield conclusions, they are composable units that ultimately lead to a final decision or answer. We synthesize a large corpus of logical chains by first collecting expert-written reasoning traces from guideline documents and QA datasets where available. For documents lacking explicit reasoning, we augment them using semi-automated paths retrieved from medical knowledge graphs or generated by prompting GPT-4 with document-specific constraints (details in Appendix A.3). To retrieve relevant paths, we train a ColBERT-based retriever and validate its quality using recall@k on expert-annotated examples and manual spot-checks (see Appendix A.4.2 for evaluation).

This yields a corpus $\mathcal{D} = \{(d_i, \mathbf{z}_i)\}_{i=1}^N$, where $d_i$ is a document (or case context) and $\mathbf{z}_i = \langle z_{i,1} \to \cdots \to z_{i,T_i} \rangle$ is its paired logical chain; the corpus contains over 2M (document, chain) pairs.

**Design Principle 2 (Entity-Guided Abstraction of Clinical Logic).** Let $\mathcal{E}$ be the universe of clinical entities, and let $\phi(d) \subseteq \mathcal{E}$ extract the entities mentioned in document $d$. For each paired example $(d_i, \mathbf{z}_i)$ we enforce *entity alignment*:

$$\mathrm{Ent}(\mathbf{z}_i) \subseteq \phi(d_i),$$

where $\mathrm{Ent}(\mathbf{z}_i)$ is the set of entities referenced across the steps in $\mathbf{z}_i$. This grounds every transition $\tau_{i,t} = (z_{i,t} \to z_{i,t+1})$ in the semantic content of $d_i$, guiding abstraction while preserving local coherence.

**Design Principle 3 (Chain-Enriched QA Construction).** Each $(d_i, \mathbf{z}_i)$ is extended with a QA pair $(q_i, g_i)$ such that the answer is *entailed* by the chain: $\mathbf{z}_i \models g_i$ (i.e., $g_i$ follows from the stepwise inferences in $\mathbf{z}_i$). We generate $(q_i, g_i)$ from $(d_i, \mathbf{z}_i)$ using domain templates, ensuring questions require the chain rather than isolated facts. These QA tuples are used (1) to train a chain retriever with triplets $(q_i, \mathbf{z}_i^+, \{\mathbf{z}_{i,k}^-\})$ and (2) to evaluate how well retrieved chains support answer generation.

## 4 TTT-RAG

Humans handle both RQ1 and RQ2 naturally: before answering, we jot down a tentative chain of steps (making the reasoning explicit, **RQ1**) and then look up sources targeted to the weakest step to confirm or fix it (aligning evidence to the chain, **RQ2**). In contrast, SLMs are pretrained to mimic surface heuristics, keep their chains latent, and retrieve by the question (or final answer) rather than by the failing step, so they neither expose where they're unsure nor fetch evidence for the exact broken link.

We operationalize the human workflow with TTT–RAG. Stage A (elicitation for RQ1). Given a test query, we retrieve a tiny minibatch of related $(Q, A, \text{chain})$ exemplars and perform lightweight test-time training so the model commits to a provisional, instance-specific chain before any external evidence is shown. Stage B (step-aligned retrieval for RQ2). We convert each transition in that chain into a focused query and retrieve step-aligned documents that supply the missing premise or surface contradictions at that link; the goal is not to hard-repair the chain, but to present exactly the evidence the model needs to revise its own step. This two-stage procedure mirrors human practice—first externalize the plan, then read precisely for the fragile parts—thereby addressing **RQ1** and **RQ2** within a single, end-to-end method.

**Stage A: Test-time elicitation via chain-alignment.** To externalize the model's hidden reasoning for the current query **(RQ1)**, we perform a brief, neighbor-conditioned *test-time chain alignment* so the model first *commits* to an explicit, step-indexed plan before any evidence is retrieved. Given a test query $q$, a chain-aware retriever $r_\phi$ (with parameters $\phi$) returns a tiny exemplar set of chain triples $\mathcal{S}(q) = \{(q^{(i)}, y^{(i)}, \mathbf{z}^{(i)})\}_{i=1}^m$ of size $m$, where each $\mathbf{z}^{(i)} = \langle z_1^{(i)} \to \cdots \to z_{T_i}^{(i)} \rangle$ is a step-indexed logical chain paired with question $q^{(i)}$ and answer $y^{(i)}$. Starting from base model parameters $\theta$, we take a few gradient steps with learning rate $\eta_{\mathrm{cot}}$ that *only* align the output distribution $\pi_\theta(\cdot)$ to explicit chains:

$$\theta_q = \theta - \eta_{\mathrm{cot}} \nabla_\theta \left( \frac{1}{m} \sum_{i=1}^m - \log \pi_\theta(\mathbf{z}^{(i)} \mid q^{(i)}) \right),$$

yielding query-adapted parameters $\theta_q$. We then elicit a provisional chain for the instance:

$$\hat{\mathbf{z}} = \langle \hat{z}_1 \to \cdots \to \hat{z}_T \rangle \sim \pi_{\theta_q}(\cdot \mid q), \qquad \hat{\tau}_t = (\hat{z}_t \to \hat{z}_{t+1}), \ t = 1, \ldots, T-1,$$

where $T$ is the length of the elicited chain and $\hat{\tau}_t$ denotes its $t$-th transition. By collapsing diffuse latent hypotheses into a concrete sequence $\hat{\mathbf{z}}$, Stage A makes intermediate states and transitions observable on this instance—fulfilling **RQ1** and furnishing the step queries $\{\hat{\tau}_t\}$ used in Stage B.

**Stage B: Step-aligned retrieval and answer generation.** To align evidence with the model's *provisional* reasoning from Stage A, we retrieve documents *by transition* rather than by the raw question. Given the elicited chain $\hat{\mathbf{z}}$ and its transitions $\hat{\tau}_t$ for $t = 1, \ldots, T-1$, a chain-aware retriever $r_\phi$ (parameters $\phi$) forms a step-conditioned query from $(q, \hat{\tau}_t)$ and returns a small, ranked set of step-aligned documents

$$\mathcal{E}_t = \mathrm{TopK}_k\big(r_\phi(q, \hat{\tau}_t)\big) \quad \text{with} \quad \mathcal{E}_{1:T} = \{\mathcal{E}_t\}_{t=1}^{T-1}.$$

Optionally, a step-level alignment scorer $s_\psi(d\,;\,q, \hat\tau_t) \in [0, 1]$ (parameters $\psi$) filters each pool by a threshold $\tau$:

$$\widetilde{\mathcal{E}}_t \;=\; \{\, d \in \mathcal{E}_t \,:\, s_\psi(d\,;\,q, \hat\tau_t) \geq \tau \,\}, \qquad \widetilde{\mathcal{E}}_{1:T} = \{\widetilde{\mathcal{E}}_t\}_{t=1}^{T-1}.$$

We condition the model on the step-aligned evidence bundle and let it give the final answer:

$$y' \;\sim\; \pi_{\theta_q}\big(\;\cdot\;\big|\, q,\, \hat{\mathbf{z}},\, \widetilde{\mathcal{E}}_{1:T}\big),$$

optionally emitting an updated chain $\mathbf{z}'$ along with $y'$. By conditioning retrieval on each transition $\hat\tau_t$, Stage B supplies precisely the missing premises or contradictions at the fragile links of the chain—thereby aligning evidence to the model's stepwise plan and addressing **RQ2**.

## 5 Experiment

### 5.1 Datasets & Models

**Datasets** We evaluate on three medical QA sets—**MedQA** (Jin et al., 2021) (USMLE Step 1/2CK/3, Jun 2022–Mar 2023), **MedMCQA** (Pal et al., 2022) (194k MCQs across 21 subjects), and **MMLU–Medical** (Hendrycks et al., 2020) (nine medical areas)—and two general multi-hop benchmarks: **2WikiMultihopQA** (Ho et al., 2020) and **MultiHopQA** (Song et al., 2018), which require cross-document evidence and step-wise verification.

**Models** Our backbone is Qwen2.5 (7B/14B). Baselines include instruction-tuned LLMs (**Qwen2.5-Instruct**), RAG systems (**MedRAG**, **i-MedRAG**), and test-time reasoning (**rStar–Qwen2.5**); for broader comparison we include open-source reasoning models (e.g., **NuminaMath**, **LLaMA3**, **Mathstral**) and proprietary LLMs (**GPT-4o**, **Claude 3.5**). TTT–RAG consistently outperforms open-source and RAG baselines across medical and general multi-hop tasks.

### 5.2 Evaluation Settings

**RQ1: Does TTT–RAG elicit high-quality logical chains?** We test chain–elicitation by directly evaluating the produced reasoning chains. Concretely, we prompt models to "think then answer" and compare a no-adaptation CoT baseline and strong open models (Qwen2.5 7B/14B/32B, Llama3.1 8B, Mistral-7B, Phi-4 14B) against TTT–RAG-14B. To further assess the quality of generated logical chains, we conduct both automatic and human evaluations. Automatically, we measure the alignment between generated chains and gold references using string similarity and step-level entailment accuracy(Atomic Cov.). For human evaluation, we randomly sample 100 logical chains from different model variants (e.g., instruction-tuned, RAG-based, logical chain-based) and ask annotators to rate them on three criteria: (1) logical coherence, (2) domain correctness. We will compare each model's logical chain with our method and calculate the win rate.

**RQ2: Does TTT–RAG correct the wrong step and improve accuracy?** We use two complementary settings. Cross–domain comparison: on the medical (MedQA, MedMCQA, MMLU–Medical) and general multi–hop suites (MultiHopQA, 2Wiki), we run the same backbones (Qwen2.5–7B/14B) and compare four method families designed to isolate confounds: (i) CoT baselines (Qwen2.5 CoT, rStar) to control for "more thinking" without retrieval; (ii) Only RAG (MedRAG) to control for document access without chain guidance; (iii) CoT+RAG (interactive) (i–MedRAG) to control for queryable evidence without step indexing; and (iv) TTT–RAG (ours), which first elicits a step–indexed chain and then converts each step into a subquery whose retrieved evidence conditions the final answer. Furthermore, within MedQA we further isolate our components with three controlled configurations that match backbone, prompts, and decoding: **Setting 1** TTT (no logical–chain data, no retrieval; direct answer), **Setting 2** TTT (Logical) (with logical–chain synthesis data, still direct answer; no retrieval), and **Setting 3** TTT–RAG (generate a structured chain, map steps to subqueries, retrieve step–aligned evidence, then answer). This design tests RQ2 mechanistically: if gains appear only when elicited chains drive step–specific retrieval (Setting 3) and persist against (i)–(iii) across domains, the accuracy lift is attributable to correcting the specific erroneous step by injecting the right premise at the right step, rather than to CoT alone, TTT alone, vanilla RAG, or interactive CoT+RAG.

# 6 RESULTS

## 6.1 RQ1: TTT–RAG ELICIT HIGH-QUALITY LOGICAL CHAINS

Relative to plain CoT baselines, test-time adaptation stabilizes reasoning but still trails chain-aware variants, as the chain–quality comparison on MedQA shows (Table 1). Among strong non-TTT open models, ROUGE-1 clusters in a narrow band (Qwen2.5-7B 0.2841 ± 0.0063, Llama3.1-8B 0.2640 ± 0.0146, Mistral-7B 0.2617 ± 0.0012, Phi-4-14B 0.2438 ± 0.0012), while

| Model | ROUGE-1 | Atomic Cov. | Human |
|---|---|---|---|
| Qwen2.5 7B | 0.2841 ± 0.0063 | 0.3541 ± 0.0063 | 31% |
| Llama3.1 8B | 0.2640 ± 0.0146 | 0.3215 ± 0.0270 | 39% |
| Mistral-7B | 0.2617 ± 0.0012 | 0.3211 ± 0.0021 | 30% |
| Phi-4 (14B) | 0.2438 ± 0.0012 | 0.3213 ± 0.0040 | 45% |
| Qwen2.5 14B | 0.2429 ± 0.0646 | 0.5437 ± 0.0682 | 41% |
| Qwen2.5 32B | 0.2225 ± 0.0003 | 0.5936 ± 0.0017 | 41% |
| TTT-RAG-14B | **0.3318** ± 0.0130 | **0.6112** ± 0.0210 | – |

Table 1: Performance of different models on reasoning quality in the MedQA dataset.

step-level entailment (*Atomic Cov.*) grows mainly with scale on the same backbone (Qwen2.5-14B 0.5437 ± 0.0682 → Qwen2.5-32B 0.5936 ± 0.0017), indicating larger models assert more facts that are actually supported without necessarily improving lexical/structural alignment to reference chains. Human ratings for these baselines (30–45%) similarly suggest fluent but heuristic chains. By contrast, our pipeline's chains (TTT-RAG-14B) achieve the strongest automatic quality—ROUGE-1 **0.3318** ± 0.0130 (absolute gain ≈ +0.048 over the best baseline, Qwen2.5-7B) and *Atomic Cov.* **0.6112** ± 0.0210 (gain ≈ +0.018 over Qwen2.5-32B)—showing both tighter alignment to gold chains and higher factual support per step.

## 6.2 RQ2: TTT–RAG CORRECT THE WRONG STEP AND IMPROVE ACCURACY

Adding step–indexed supervision during TTT yields a modest but consistent gain when the model answers without retrieval (TTT (Logical) in Fig. 3 Top: 55.6 → 58.7 on MedQA; consistent with the small Only TTT → TTT+CoT lift in Table 3). This indicates that supervision helps the model articulate cleaner intermediate premises, but it does not correct wrong steps caused by missing facts: without evidence injection, a mistaken premise remains unverified and the final answer often stays wrong. The correction appears only when the elicited chain drives retrieval: mapping each step to a subquery boosts MedQA to 70.1 for TTT–RAG in Fig. 3, and the same mechanism scales across domains and sizes in Table 2 (e.g., TTT–RAG–14B = 73.8/64.3 on MedQA/MedMCQA and 48.5/44.6 on MultiHopQA/2Wiki, surpassing CoT, rStar, MedRAG, and i–MedRAG). These controlled comparisons explain **why competing methods fall short**: CoT/rStar provide longer reasoning but cannot inject missing facts; MedRAG re-

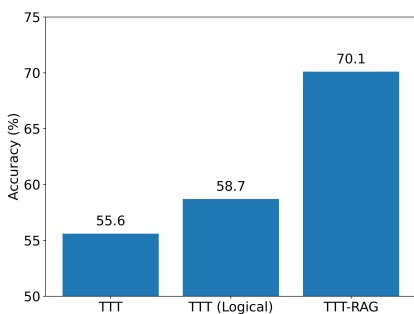

Figure 3: Ablation study analyzing two design choices in TTT-RAG: (Top) the proportion of logical chains retained during test-time training, and (Bottom) the number of parallel users processed in batch-wise adaptation.

trieves at the question level and often misaligns evidence with the needed step; and i–MedRAG issues ad-hoc queries without explicit step IDs, so overconfidence can suppress the very query required. In contrast, TTT–RAG uses elicited, step-indexed chains to localize the failing link and fetch the right premise at the right step, yielding the observed accuracy gains.

## 6.3 EFFICIENCY

The table 3 disentangles the contributions of CoT, RAG, and TTT to accuracy and latency. Pure CoT (55.1, +0s) is the weakest. Adding more "thinking" without retrieval via rStar yields only a small gain (58.1, +497s), indicating poor accuracy–latency efficiency (≈ +3.0/497). Either component alone helps modestly: Only RAG improves to 58.7 at no extra time (+3.6, +0s), while Only TTT reaches 59.1 with moderate overhead (+4.0, +24s). Combining CoT with generic interactive RAG (iMedRAG) produces a larger boost (62.5, +26s), showing synergy between reasoning and document access, yet accuracy remains limited because retrieval is not aligned to specific steps. Introducing TTT on top of CoT but without retrieval (TTT+CoT) barely moves the needle (59.3, +25s), implying that elicited

| | Medical Domain | | | General Domain | |
|---|---|---|---|---|---|
| Model | MedQA | MedMCQA | MMLU | MultiHopQA | 2Wiki |
| Qwen2.5-7B-Instruct | 53.2 | 54.1 | 70.3 | 26.3 | 27.4 |
| Qwen2.5-14B-Instruct | 60.8 | 55.7 | 75.2 | 32.5 | 30.1 |
| MedRAG-Qwen2.5-7B | 51.0 | 56.9 | 68.1 | 33.2 | 31.2 |
| i-MedRAG-Qwen2.5-7B | 54.3 | 57.6 | 74.4 | 35.6 | 34.4 |
| rStar-Qwen2.5-7B | 58.1 | 55.9 | 69.8 | 29.1 | 28.5 |
| rStar-Qwen2.5-14B | 63.2 | 56.7 | 77.2 | 34.1 | 30.6 |
| TTT-RAG-7B | 70.1 | 62.1 | 75.4 | 45.1 | 42.8 |
| TTT-RAG-14B | **73.8** | **64.3** | **78.5** | **48.5** | **44.6** |

Table 2: Model Performance split by *Medical* vs *General* domains.

| Method | Uses CoT | Uses RAG | Uses TTT | Accuracy (MedQA) | Time / Q (s) |
|---|---|---|---|---|---|
| Qwen-2.5 (CoT) | ✓ | ✗ | ✗ | 55.1 | +0s |
| Qwen-2.5-rStar | ✓ | ✗ | ✗ | 58.1 | +497s |
| Only TTT | ✗ | ✗ | ✓ | 59.1 | +24s |
| Only RAG (MedRAG) | ✗ | ✓ | ✗ | 58.7 | +0s |
| CoT+RAG (iMedRAG) | ✓ | ✓ | ✗ | 62.5 | +26s |
| TTT+CoT | ✓ | ✗ | ✓ | 59.3 | +25s |
| TTT+CoT+RAG | ✓ | ✓ | ✓ | 70.1 | +32s |

Table 3: Component usage, accuracy, and latency on MedQA (Figure 2). *Time / Q* reports additional per-question latency relative to the Qwen-2.5 (CoT) baseline.

chains alone do not reliably correct mistakes caused by missing facts. The full TTT+CoT+RAG is the clear winner (70.1, +32s): relative to iMedRAG, it delivers +7.6 points for only +6s additional latency, and relative to Only RAG it adds +11.4 points with acceptable overhead. These contrasts isolate the active ingredient: converting elicited, step-indexed chains into subqueries (TTT–RAG) enables step-synchronized retrieval that fixes the failing link, whereas "more CoT," TTT alone, or question-level/interactive RAG cannot consistently target and repair the erroneous step.

## 6.4 ROBUSTNESS: EVALUATION ON MATH DOMAIN.

For mathematical reasoning, we conduct a comprehensive evaluation on four widely-used benchmarks: MATH Lightman et al. (2023), AMC 2023 (Team, 2024), Olympiad Bench He et al. (2024), and GSM8K Cobbe et al. (2021). We conduct further ablation studies to better understand the behavior of our TTT-RAG framework:

**Impact of Small-Batch Size during Test-Time Training.** We systematically vary the size of the adaptation batch used in TTT (e.g., 1, 2, 4, 8 examples) to assess how the quantity of retrieved supervision affects the quality of adaptation and final accuracy. This ablation helps reveal the trade-off between adaptation speed and generalization performance.

**Parallel Multi-Query Test-Time Adaptation.** To simulate a multi-user inference setting and evaluate scalability, we group concurrent user queries (e.g., 3, 6, 9, 12 queries) and perform joint adaptation using a concatenated retrieval batch. This parallel strategy aims to amortize the cost of TTT while enabling more efficient resource sharing. We measure how this affects accuracy and adaptation stability across different parallelism levels.

**Ablation Study: Mathematical Domain.** To investigate how different configurations of logical chain integration

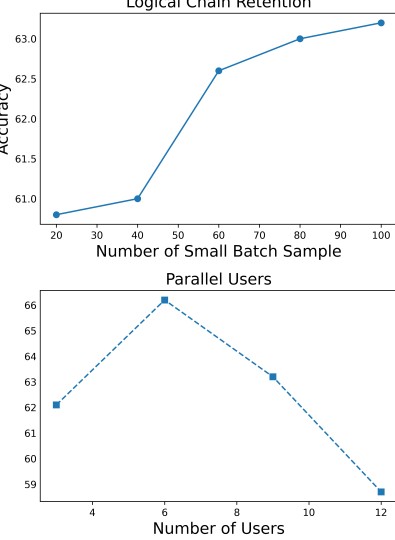

Figure 4: Ablation study for TTT-RAG. (Top) Effect of the number of retained small-batch samples used in logical chain supervision. (Bottom) Impact of varying the number of parallel users during batch-wise test-time training.

and test-time parallelization affect mathematical reasoning performance, we conduct two sets of ablation studies, visualized in Table 4.

In the Figure 4, we vary the **retention percentage** of retrieved logical chains during test-time training—from 20 to 100 samples and observe a monotonic increase in accuracy. This indicates that maintaining a greater proportion of logical reasoning steps during adaptation substantially benefits the model's ability to solve complex symbolic problems, improving accuracy from 60.9% to 63.2%. We also evaluate the effect of parallel user batch size, where retrieved batches from 3, 6, 9, or 12 users are aggregated and jointly used for test-time training. We find that moderate parallelism (6 users) improves performance by reducing variance in reasoning paths, while excessive parallelism (12 users) introduces noise and leads to degraded accuracy. This suggests a trade-off between parallel efficiency and reasoning fidelity, and highlights the importance of controlled batch-level adaptation for symbolic reasoning tasks.

# 7 RELATED WORK

**Reasoning with LLMs and RAG.** LLMs excel broadly yet still struggle with explicit, multi-step inference and modular logic (Liu et al., 2023; Wu et al., 2024; Dziri et al., 2023). Prompting (CoT, self-consistency) improves fluency but not verifiability or coherence in expert domains (Wei et al., 2022; Wang et al., 2022). RAG injects external knowledge (Lewis et al., 2020; Guu et al., 2020), but retrieved text is often unstructured or only loosely related to the *step* that needs support (Ye et al., 2023; Xiong et al., 2024a); even structured variants (e.g., MedRAG, iMedRAG) reduce hallucinations but still misalign with step-wise reasoning. We address this gap with *LogicalChain*—a retrieval-ready corpus of interpretable, step-indexed derivations (e.g., cause $\rightarrow$ mechanism $\rightarrow$ effect) that equips LLMs with structured scaffolds rather than heuristic completions.

**Test-Time Training and Adaptation.** TTT adapts models *during inference* using test-specific signals; while effective in vision for domain robustness (Sun et al., 2020; Behrouz et al., 2024), its use for NLP—especially LLM reasoning—remains limited (Lee et al.), and prior work largely relies on unsupervised objectives (e.g., shift detection, contrastive reconstruction) with little task structure or semantic alignment. In reasoning settings, we observe persistent *in-stance–step misalignment*: chains retrieved by pretrained retrievers often reflect corpus-level relevance

| Model | MATH | AMC2023 | Olympiad | GSM8K |
|---|---|---|---|---|
| Qwen2-7B-Instruct | 49.6 | 25.0 | 0.04 | 82.3 |
| NuminaMath-7B-COT | 55.8 | 27.5 | 0.03 | 76.3 |
| rStar-Qwen2 | 60.4 | 30.0 | 20.0 | **87.2** |
| **TTT-RAGs** | **63.2** | **30.0** | **24.7** | 87.0 |

Table 4: Comparison of model performance across mathematical benchmarks. rStar-Math shows strong performance across all metrics.

rather than the step-specific trajectory needed for a given case. We therefore combine TTT with RAG, adapting the model at test time via *structured* supervision on retrieved logical chains and supporting documents so that reasoning focuses on the current instance without global parameter updates. To our knowledge, this is the first approach to inject *instance-specific logical supervision* into a TTT pipeline for LLMs, enabling more robust and interpretable reasoning across medical and general multi-hop domains.

# 8 CONCLUSION

We present LOGICALCHAIN + TTT-RAG, a retrieval-augmented reasoning framework that enhances LLM performance by integrating structured logical chains through test-time adaptation. By dynamically aligning the model's reasoning process with retrieved inference paths, TTT-RAG significantly enhances performance on medical and mathematical benchmarks, surpassing existing retrieval-augmented and instruction-tuned baselines. Experimental results show that incorporating structured reasoning pathways during inference improves both the accuracy and logical coherence of model outputs. These findings demonstrate the effectiveness of logic-guided, instance-specific adaptation in advancing the reasoning capabilities of LLMs for complex tasks.

## 9 REPRODUCIBILITY STATEMENT

We have taken several steps to ensure the reproducibility of our work. All models and algorithms are described in detail in the main text (Sections 3 and 4). The description of datasets and preprocessing steps is given in Section 3 and Appendix. Hyperparameters and training configurations are reported in Section 5.

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

# A    APPENDIX

## A.1    LLM USAGE

In accordance with the ICLR 2026 policies on LLM usage, we disclose how LLMs were used in this work. LLMs were employed to assist with grammar polishing, wording improvements, and drafting text during paper preparation. All technical content, proofs, experiments, and analyses were conceived, implemented, and validated by the authors. Authors remain fully responsible for the correctness of the claims and results.

No LLMs were used to generate research ideas or produce results. No confidential information was shared with LLMs, and no prompt injections or other inappropriate uses were involved.

This disclosure aligns with the ICLR Code of Ethics: contributions of tools are acknowledged, while accountability and verification rest entirely with the human authors.

## A.2    ERROR ANALYSIS

To better understand the limitations of different retrieval strategies, we conducted an error analysis comparing a baseline RAG approach with our proposed TTT-RAG framework. Using Qwen2.5-14B as the base model, we retrieve documents and directly generate answers, and compared this with TTT-RAG's test-time adapted retrieval and inference. We randomly sampled 100 test cases and manually categorized each incorrect prediction into one or more of the following four error types: (1) irrelevant or missing evidence, (2) misleading evidence, (3) insufficient reasoning ability, and (4) forgotten knowledge.

The ColBERT-RAG setup produced 15 errors due to irrelevant or missing documents, 6 from misleading content, 20 due to reasoning failures, and 6 involving forgotten background knowledge. In contrast, TTT-RAG reduced these numbers to 7, 0, 12, and 5 respectively. These results highlight that TTT-RAG not only improves retrieval relevance but also reduces reasoning and knowledge-related errors, suggesting that test-time adaptation provides more reliable and contextually aligned inputs for downstream inference.

## A.3    DATA CONSTRUCTION PIPELINE

We categorize documents into two types: (1) **Expert-authored documents with explicit reasoning traces**, such as clinical guidelines, medical board exam explanations, or textbook problem solutions. These documents often contain structured logic that can be extracted or lightly reformatted to form usable reasoning chains. (2) **Factual documents without explicit reasoning**, including textbook paragraphs, curated QA datasets, and encyclopedic resources such as Wikipedia. These texts provide rich medical knowledge but require additional processing to reveal the underlying reasoning structure. The details are shown in Table 5

For type (2), we synthesize logical chains by prompting GPT-4 with a structured instruction template. Each input includes a factual medical passage, and the model is asked to generate a step-by-step logical chain that connects context to conclusion while avoiding large logical jumps. We include constraints to ensure that:

- The logical chain begins with the key facts or conditions in the document;

- Intermediate reasoning steps are included (e.g., physiological mechanisms, diagnostic steps, or treatment justifications);

- The final step logically derives the medically relevant conclusion (e.g., diagnosis, mechanism, treatment);

- Language remains generalizable and concise, avoiding unnecessary specifics (e.g., age, lab values) unless clinically necessary.

A representative prompt looks like this:

```
"The text below contains a medical fact. Your task is to generate a logical
chain that explains the underlying reasoning. Be concise but complete. Return
format: Logical Chain: A → B → C"
```

This process allows us to convert static medical knowledge into dynamic reasoning trajectories that can be retrieved and aligned at inference time.

### A.4 RETRIEVER TRAINING AND EVALUATION

#### A.4.1 LOGICAL CHAIN RETRIEVER TRAINING

We retrieve logical chains using a two-stage retrieval–reranking pipeline . First, we use ColBERT to retrieve the top-K candidate logical chains for each query $q$. Then, a cross-encoder is applied to rerank these candidates. This pipeline balances retrieval efficiency with reranking effectiveness, leveraging ColBERT's fast dense retrieval and the cross-encoder's stronger semantic matching capabilities. Specifically, the cross encoder is trained to map a query $q$ (typically a user question) to its corresponding logical chain $l^+$, sampled from a shared document $d$. Each training instance is constructed as a triplet $(q, l^+, \{l_i^-\}_{i=1}^N)$, where $l^+$ is the gold reasoning path and $l_i^-$ are hard negatives sampled from neighboring chains within the same document $d$. The training objective minimizes the following contrastive loss:

$$\mathcal{L} = -\log \frac{\exp(\text{Sim}(q, l^+))}{\exp(\text{Sim}(q, l^+)) + \sum_{i=1}^{N} \exp(\text{Sim}(q, l_i^-))}, \tag{1}$$

where $\text{Sim}(q, l)$ denotes the similarity score between query $q$ and candidate logical chain $l$, computed via a dual-encoder architecture. The softmax denominator encourages the model to assign the highest score to the gold chain.

Each query $q$ is associated with a unique logical chain $l$, such that the retrieved logical chain $l^+$ implies both the supporting document and the reasoning steps required to answer $q$. This alignment enables the retriever to support dual downstream tasks. QA-to-Chain Retrieval: Given a clinical question $q$, retrieve the corresponding logical chain $l^+$ that reflects guideline-based reasoning. Guideline-to-Chain Retrieval: Given a document excerpt $d$, retrieve an appropriate reasoning path $l^+$ for diagnostic or therapeutic inference. By training jointly on both tasks, the retriever learns generalizable alignment between questions, reasoning chains, and medical documents, supporting structured retrieval across diverse input scenarios.

#### A.4.2 EVALUATON RESULTS

We compare the retrieval accuracy of our proposed Logical Chain Retriever against a strong baseline, ColBERT Retriever, on top-$k$ evidence selection. For the test datasets, we construct QA pairs based on Pubmed corpus to test. As shown in Table 6, the Logical Chain Retriever consistently outperforms ColBERT across all values of $k$.

| | | | | |
|---|---|---|---|---|
| Logical Chain Retriever | 73.27 | 77.49 | 80.13 | 81.79 |
| ColBERT Retriever | 62.33 | 65.12 | 67.48 | 70.51 |
| Logical Chain Retriever | 73.27 | 77.49 | 80.13 | 81.79 |
| ColBERT Retriever | 62.33 | 65.12 | 67.48 | 70.51 |

Table 6: Top-$k$ retrieval accuracy (%) comparison between Logical Chain Retriever and ColBERT Retriever.

Notably, it achieves 73.27% top-5 accuracy, compared to 62.33% with ColBERT. This performance

Table 5: Prompt Template for Logical Rule Generation.

| | |
|---|---|
| **System Prompt** | The text below contains medical information. Your task is to create rules based on the medical knowledge presented, ensuring each rule comprehensively represents the information in the evidence text without omitting any significant details. |
| **1. Clear Distinction Between Conditions and Outcomes** | Each rule should clearly reflect when certain items are conditions, patient characteristics, or prerequisites (such as specific diagnoses, clinical findings, or circumstances) and when others are outcomes, actions, or recommendations. Always list conditions at the start of the rule, followed by the recommended actions or results. |
| **2. Multiple Rules for Complex Evidence** | If an evidence statement includes multiple distinct aspects, create separate rules for each part. This ensures that each concept is represented fully and clearly, preventing unrelated ideas from being merged into a single rule. Reuse the evidence as needed for each distinct rule. |
| **3. Parallel Items and Logical Structure** | Arrange items logically within each rule. 
• For items that have a natural sequence based on real-world logic (e.g., patient characteristics → diagnostics → treatment recommendations), follow this order. 
• If items are parallel without a strict sequence, indicate them at the same level within the rule without imposing order (e.g., "PET benefits → Diagnosis, Staging, Treatment Response"). |
| **4. Real-World Logical Order** | When applicable, follow a natural sequence based on real-world relationships rather than sentence order. The preferred structure is: 
• Conditions or Requirements 
• Diagnostics 
• Findings or Results 
• Treatment Recommendations or Actions 
• Follow-Up or Monitoring Plans (if applicable) 
Present these elements in this logical order, regardless of how they are arranged in the original text. |
| **5. Reconstruct Guideline** | Ensure that the generated set of evidence and rules covers the guideline comprehensively, allowing the guideline to be reconstructed accurately from these parts. Use exact phrasing from the original guideline text where feasible. |
| **6. Include Every Key Detail** | Represent every significant term, phrase, or concept from the evidence. Each rule should be as concise as possible while fully representing all critical details. Avoid redundancy, but include all essential information. |
| **7. Direct Language Without Conditionals** | Use clear and direct language without excessive use of conditionals like "if" or "when." Each rule should be logically complete and self-contained, conveying the full context of the information. |
| **8. Output Format Medical Text**: 
{medical_text} 

**Example Rule Format:** | **Key Terms to Include**: {key_words} |
| | **Correct:** 
**Evidence**: "A patient presents with abdominal pain, followed by a CT scan, which shows an abscess. Treatment includes antibiotics." 
**Rule**: Abdominal pain → CT scan → Abscess detected → Treatment: Antibiotics. 
**Incorrect:** 
**Evidence**: "A patient presents with abdominal pain, followed by a CT scan, which shows an abscess. Treatment includes antibiotics." 
**Rule**: Abdominal pain → Abscess detected → CT scan → Treatment: Antibiotics. |
| **Return Format** | • Evidence: 
• Rule: 
• Evidence: 
• Rule: 
Return only the specified output format without additional commentary or explanation. |

gap widens with larger $k$, reaching 81.79% at top-20 retrieval, demonstrating the effectiveness of incorporating structured reasoning signals during retrieval. These results indicate that explicitly mod-

eling logical chains not only improves alignment with the query intent, but also facilitates retrieval of more relevant and explanatory evidence.

## A.5 QA GENERATION EXAMPLES

Given a logical chain, we automatically generate QA pairs by framing the final step as the answer and rephrasing the earlier reasoning path as a question. This alignment ensures that the question is logically entailed by the steps in the chain, and that the answer is the natural conclusion of the reasoning process.

For example, given the input:

**Document**: "Cisplatin binds DNA and induces cross-linking, which causes apoptosis."
**Logical chain**: "Cisplatin → DNA crosslinking → blocks replication → apoptosis"

We generate:

**Q**: What is the therapeutic mechanism of Cisplatin?
**A**: DNA cross-linking

This question-answer pair reflects both the factual content and the reasoning structure, enabling it to be used in alignment tasks or as a target for supervised fine-tuning. When needed, we apply controlled variation (e.g., passive voice, question style) to increase linguistic diversity while preserving the underlying logic.

## A.6 EVALUATION PROMPT DESIGN

When evaluating close QA, we only need to calculate accuracy. However, many open QA tasks, such as diagnostic reasoning questions in the Amboss Dataset, present additional challenges. Although several methods exist for measuring textual similarity, such as F1 or ROUGE, both approaches have significant limitations in the medical domain. Therefore, we propose a very strict evaluation pipeine by using two evaluation metrics: the USMLE-Factuality score and the GPT-4o score. For the GPT-4o score, directly allowing GPT-4o to grade the answers is often ineffective, as GPT-4o tends to favor answers that align with its preferred linguistic style, which may not match our intended criteria. Thus, we introduce a scoring framework to evaluate model's fine grained diagnostic ability based on three aspects: **Key Points**, **Inference**, and **Evidence** which is designed by doctors:

- **Key Points** assess whether the model's answer includes the critical elements present in the ground truth.
- **Inference** evaluates whether the diagnostic reasoning in the model's answer is correct, follows the same steps as the ground truth, and whether any key steps are omitted.
- **Evidence** examines whether the model's answer provides the crucial evidence to support its conclusions or diagnostic reasoning.

Finally, an average score will be calculated to represent the overall quality of the answer. To further reduce the influence of linguistic style on GPT-4's scoring, we propose revising all model-generated answers through GPT-4, ensuring that all outputs align with GPT-4's own style distribution. During this revision, GPT-4 will only see the model's answer, without access to any other information.

When scoring, GPT-4 will generate its own summaries of **Key Points**, **Inference**, and **Evidence** based on the ground truth. When assigning scores to these aspects, GPT-4 will no longer see the original answer but will only reference its summarized **Key Points**, **Inference**, and **Evidence**.

Table 7: Evaluation Pipeline Prompt Example Template.

| | |
|---|---|
| **System Prompt** | The text below contains medical information. Your task is to create rules based on the medical knowledge presented, ensuring each rule comprehensively represents the information in the evidence text without omitting any significant details. |
| **Clear Distinction** | Each rule should clearly reflect when certain items are conditions, patient characteristics, or prerequisites (such as specific diagnoses, clinical findings, or circumstances) and when others are outcomes, actions, or recommendations. Always list conditions at the start of the rule, followed by the recommended actions or results. |
| **Multiple Rules for Complex Evidence** | If an evidence statement includes multiple distinct aspects, create separate rules for each part. This ensures that each concept is represented fully and clearly, preventing unrelated ideas from being merged into a single rule. Reuse the evidence as needed for each distinct rule. |
| **Parallel Items and Logical Structure** | Arrange items logically within each rule. - For items that have a natural sequence based on real-world logic (e.g., patient characteristics -> diagnostics -> treatment recommendations), follow this order. - If items are parallel without a strict sequence, indicate them at the same level within the rule without imposing order, such as "PET benefits -> Diagnosis, Staging, Treatment Response" if they all apply equally. |
| **Real-World Logical Order** | Over Sentence Structure**: When applicable, follow a natural sequence based on real-world relationships rather than sentence order. The preferred sequence generally follows: - **Conditions or Requirements** (e.g., patient characteristics or specific criteria) - **Diagnostics** (e.g., tests performed) - **Findings or Results** (e.g., outcomes of diagnostics) - **Treatment Recommendations or Actions** (e.g., proposed treatments based on findings) - **Follow-Up or Monitoring Plans** (if applicable) Present these elements in this logical order, regardless of how they are arranged in the original text. |
| **Reconstruct Guideline** | Ensure that the generated set of evidence and rules covers the guideline comprehensively, allowing the guideline to be reconstructed accurately from these parts. Each evidence should be presented as close to the original guideline text as possible, using exact phrasing where feasible. |
| **Include Details** | Represent every significant term, phrase, or concept from the evidence. Each rule item should be as concise as possible while fully representing all critical details. Avoid unnecessary redundancy but ensure all essential information is included. Use clear and direct language without excessive use of conditionals like "if" or "when." Each rule should be logically complete and self-contained, conveying the full context of the information. |
| **Output Format** | **Key Terms to Include**: key words **Medical Text**: medical text **Medical Path**: medical path **Example Rule Format**: - **Correct**: - **Evidence**: "A patient presents with abdominal pain, followed by a CT scan, which shows an abscess. Treatment includes antibiotics." - **Rule**: Abdominal pain -> CT scan -> Abscess detected -> Treatment: Antibiotics. - **Incorrect** (based purely on sentence order): - **Evidence**: "A patient presents with abdominal pain, followed by a CT scan, which shows an abscess. Treatment includes antibiotics." - **Rule**: Abdominal pain -> Abscess detected -> CT scan -> Treatment: Antibiotics. **Return Format**: 1. Evidence: Rule: 2. Evidence: Rule: Return only the specified output format without additional commentary or explanation. |

## B  CASE STUDY

Table 8: Evaluation Pipeline Prompt Example Template.

| | |
|---|---|
| **Extract Key Points** | Based on the question and answer, summarize ten key points that you consider to be the most crucial from the standard answer. Return the response in the following format: {1.2.3....} Here is the question:{question} Here is the answer:{answer} Please do not provide any additional information. |
| **Key Points** | 1. Multifocal electroretinogram (ERG) showed reduced signal in the right eye throughout the macula, confirming the diagnosis of AZOOR.2. Acute zonal occult outer retinopathy (AZOOR) was first described by Gass in 1993... |
| **Extract Diagnostic Reasoning** | Based on the question and answer, please provide a detailed summary of the diagnostic reasoning from the standard answer. Return the response in the following format: {1.2.3....} Here is the question:{question} Here is the answer:{answer} Please do not provide any additional information. |
| **Diagnostic Reasoning** | 1. The patient is a 7-year-old boy with a slowly growing, asymptomatic lump on the left lower neck since birth.2. Physical examination showed a yellowish, hump-like mass with a hairy surface and cartilage-like consistency near the left sternocleidomastoid muscle... |
| **Extract Evidence** | Based on the question and answer, please provide a detailed evidence list which is proposed by correct answer. Return the response in the following format: {1.2.3....} Here is the question:{question} Here is the answer:{answer} Please do not provide any additional information. |
| **Evidence** | 1. Slowly growing, asymptomatic lump on left lower neck since birth.2. Physical examination revealed a yellowish, hump-like mass with hairy surface and cartilage-like consistency.3. Ultrasonography indicated a hypoechoic, avascular, bulging nodule with an anechoic tubular structure.4. MRI demonstrated a protuberant nodule with diffuse... |
| **Key Points Score** | Act as a USMLE evaluator, your role involves assessing and comparing a medical student's explanation to the provided target answer. Begin the assessment by carefully reviewing the provided target answer. Then, based on following specific criteria, determine the score for the student's answer. Please judge whether medical student's answer include these key points(or some other relevant points. But the amount of points must be complete). For example, ground truth have 10 key points, if student answer include one key he will get 0.5 point(if the answer include 5 points so should be 2.5). Medical student's answer: {answer} Key Points: {Key Point} Please only return a float number(from 0 to 5). You should check each point one by one(shouldn't judge based on language style such as fluence and so on. Only judge based on whether the student's answer include correct or relevant and complete key points). Don't generate any other information. |

Table 9: Sources of Medical Guidelines, Information, and Corpus Statistics

| Source/Corpus | Full Name | Guidelines/#Doc. | Words/#Snippets | Audience/Domain | Country | Released |
|---|---|---|---|---|---|---|
| CCO | Cancer Care Ontario | 87 | 199K | Doctor | Canada | Yes |
| CDC | Center for Disease Control and Prevention | 621 | 6.7M | Doctor | USA | Yes |
| CMA | Canadian Medical Association | 431 | 1.7M | Doctor | Canada | Yes |
| ICRC | International Committee of the Red Cross | 49 | 1.2M | Doctor | International | Yes |
| NICE | National Institute for Health and Care Excellence | 1656 | 8.1M | Doctor | UK | Yes |
| PubMed | PubMed | 1627 | 10.8M | Doctor | Mix | Yes |
| SPOR | Strategy for Patient-Oriented Research | 217 | 1.1M | Doctor | Canada | Yes |
| WHO | World Health Organization | 223 | 3.1M | Both | International | Yes |
| WikiDoc | WikiDoc | 33058 | 34M | Both | International | Yes |
| StatPearls | StatPearls | 9.3k | 301.2k | Both | International | Yes |
| Textbooks | Textbooks | 18 | 125.8k | Both | International | Yes |
| Wikipedia | Wikipedia | 6.5M | 29.9M | Both | International | Yes |
| MedCorp | MedCorp | 30.4M | 54.2M | Both | International | Yes |

Table 10: Human Annotated Sample Case.

**Question:** A 67-year-old man with transitional cell carcinoma of the bladder comes to the physician because of a 2-day history of ringing sensation in his ear. He received this first course of neoadjuvant chemotherapy 1 week ago. Pure tone audiometry shows a sensorineural hearing loss of 45 dB. The expected beneficial effect of the drug that caused this patient's symptoms is most likely due to which of the following actions? A: Inhibition of proteasome, B: Hyperstabilization of microtubules, C:Generation of free radicals, D: Cross-linking of DNA

**Phi-4:** To determine the expected beneficial effect of the drug that caused the patient's symptoms, we need to identify the drug likely responsible for the sensorineural hearing loss and understand its mechanism of action. 1. The patient has transitional cell carcinoma of the bladder and received neoadjuvant chemotherapy. 2. A common chemotherapy drug used for bladder cancer is cisplatin. 3. Cisplatin is known to cause ototoxicity, which can manifest as sensorineural hearing loss. 4 Cisplatin will lead generation of free radicals and lead hearing loss hence the final answer should be C.

**Qwen-2.5-14B-Instruct** Forms of chemotherapy used in transitional cell carcinoma of the bladder often include platinum-based agents like cisplatin. Cisplatin is known for its ototoxicity, which can lead to sensorineural hearing loss. -> Cisplatin's mechanism of action involves the generation of free radicals, which can damage cellular components including those in the auditory system. -> Therefore, the generation of free radicals is the most likely action causing the patient's symptoms. -> The correct answer is C

**TTT-RAG:**

**Small Batch Finetune:**

**Pretrain Data:** MDMA: 3,4-Methylenedioxymethamphetamine (MDMA), commonly known as ecstasy (E) or molly, is a psychoactive drug primarily used for recreational purposes. The desired effects include altered sensations, increased energy, empathy, as well as pleasure. When taken by mouth, effects begin in 30 to 45 minutes and last 3 to 6 hours...

**Finetune Data:** Scenario: Dr. Smith, a seasoned urologist, is preparing for a surgical procedure to repair an anterior urethral stricture in a 45-year-old male patient. The patient is informed that a surgical resident will be involved in the operation as part of their training. The patient expresses concern about the potential impact of the resident's involvement on the surgery's...

**Inference:** 1. Transitional cell carcinoma of the bladder -> Neoadjuvant chemotherapy -> Agent causing ototoxicity leading to sensorineural hearing loss -> Correct action of the drug is hyperstabilization of microtubules 2. Transitional cell carcinoma of the bladder -> Neoadjuvant chemotherapy -> Ototoxicity side-effect -> Sensorineural hearing loss -> Caused by generation of free radicals 3.Transitional cell carcinoma of the bladder -> Neoadjuvant chemotherapy -> mechanism is Cross DNA Link -> Chemotherapeutic agent causing ototoxicity -> Sensorineural hearing loss observed -> Ototoxic mechanism involves generation of free radicals -> Answer: E

**Retrieve to Verify:** 1. Cisplatin is a chemotherapy drug → causes ototoxicity (hearing loss) → due to accumulation of reactive oxygen species (ROS) in the cochlea → ROS cause oxidative stress and damage cochlear hair cells → Antioxidants neutralize ROS and reduce cochlear damage → therefore, antioxidants can help prevent cisplatin-induced hearing loss 2. Chemotherapy Drug -> Cross DNA Link -> Therapy

**Final Answer:** D

**Doctor Comment:**The clinical reasoning task requires distinguishing between a drug's therapeutic mechanism and its side-effect profile. While models like Phi-4 and Qwen-2.5-14B correctly identified cisplatin as the causative agent, they misattributed the drug's mechanism of action to its ototoxic side effect (free radical generation). In contrast, TTT-RAG correctly disambiguates the therapeutic mechanism (DNA cross-linking) from side effects, demonstrating a structured reasoning pathway that mirrors human clinical logic.

