# OpenReview forum: "Enhancing LLM Reasoning with Retrieval-Augmented Logical Chains and Test-Time Adaptation"
_ICLR.cc/2026/Conference — Submitted to ICLR 2026_

### Official Review · Reviewer_72Si · 2025-10-29

**Soundness:** 2
**Presentation:** 2
**Contribution:** 2
**Rating:** 4
**Confidence:** 4

**Summary:**

This paper introduces LogicalChain with TTT-RAG, a framework designed to enhance large language model reasoning on complex multi-step tasks by combining structured logical chains with test-time adaptation. The authors construct a corpus of interpretable, step-by-step derivations from domain-rich sources and train a contrastive retriever to fetch task-relevant inference paths.  At test time, they propose TTT-RAG which first performs lightweight fine-tuning on retrieved chain exemplars to elicit a provisional reasoning chain, then retrieves step-aligned documents to verify and repair each reasoning step. The framework is evaluated primarily on medical QA benchmarks (MedQA, MedMCQA) and general multi-hop reasoning datasets (MultiHopQA, 2WikiMultiHopQA), showing substantial improvements over baselines including MedRAG, i-MedRAG, and rStar. On MedQA, TTT-RAG with Qwen2.5-7B achieves 70.1% accuracy (up from 53.8% baseline) and on 2WikiMultiHopQA reaches 42.8% accuracy for the 7B model and 44.6% for the 14B model.

**Strengths:**

The paper addresses a genuine limitation in current RAG systems by highlighting that retrieval conditioned only on the global query often misses the specific failing step in multi-hop reasoning, which is a valid observation grounded in empirical evidence. The experimental scope is reasonably comprehensive, covering both medical domain-specific benchmarks and general multi-hop QA datasets, which demonstrates some generalizability of the approach.

**Weaknesses:**

The related work section significantly underrepresents recent step-wise and iterative RAG methods that directly address the same problem space. Most critically, R3-RAG (2025) which uses reinforcement learning to teach LLMs step-by-step reasoning and retrieval achieves substantially better results on 2WikiMultiHopQA with Qwen2.5-7B reaching 62.3% accuracy compared to this work's 42.8%, and Llama-3.1-8B achieving 61.0% on the same benchmark. The paper also omits discussion of other highly relevant approaches including FAIR-RAG which introduces structured evidence assessment with iterative refinement cycles. Some other notable algorithms that were not considered as baseline include DRAGIN, which offers adaptable dynamic test time decision on retrieval and content.

**Questions:**

1- How does TTT-RAG compare directly to R3-RAG on the same benchmark splits and evaluation metrics
2-What are the wall-clock latency costs per query for TTT-RAG compared to R3-RAG and other step-wise iterative methods, not just token counts, since test-time fine-tuning introduces additional computational overhead that may not be reflected in token usage alone?

---

### Official Review · Reviewer_Wg6F · 2025-11-01

**Soundness:** 2
**Presentation:** 3
**Contribution:** 3
**Rating:** 2
**Confidence:** 4

**Summary:**

This paper proposes LogicalChain and TTT-RAG, a framework that combines retrieval-augmented generation with test-time training to improve LLM reasoning on complex, multi-step tasks. The key contributions are: (1) construction of a large corpus of structured logical chains from domain-rich sources, (2) a contrastive retriever trained to fetch task-relevant inference paths, and (3) TTT-RAG, which performs test-time adaptation by fine-tuning the LLM on retrieved chains during inference. The authors evaluate the proposed methods on medical benchmarks (MedQA, MedMCQA) and general multi-hop reasoning tasks (MultiHopQA, 2Wiki) and show substantial improvements over baselines like MedRAG and rStar.

**Strengths:**

1. The paper clearly identifies the challenge that LLMs keep reasoning chains latent, and RAG systems retrieve at the question level rather than the step level.
2. The combination of structured logical chains + test-time training + step-indexed retrieval is novel.
3. The paper shows consistent and substantial improvements across multiple medical and general-domain benchmarks.

**Weaknesses:**

1. The comparison in the experiments might not be fair. The open-source reasoning models are all trained on math reasoning, while the benchmarks are in medical and general domain. The domain gap might be the reason why these models suffer.
2. The method adds significant test-time cost. Test-time training on every query adds +32s per question (Table 3), which is not practical considering its performance gain.
3. Since the authors already created a reasoning dataset, a better set of baselines should use these data to perform SFT, RLVR, or in-context learning, which incurs much lower online computation cost.
4. There is no systematic evaluation of synthesized chain quality. What if GPT-4 hallucinates reasoning steps? How many of 2M+ chains were validated?

**Questions:**

1. How does TTT compare with traditional SFT, which adds nearly no inference-time costs? Would MedRAG/i-MedRAG improve if trained on the same generated logical chain corpus?
2. How does the method compare with RLVR-trained methods or using in-context learning instead of TTT?
2. MedQA should be a multiple-choice dataset. Why report ROUGE-1 and AtomicCov?
3. What does Compare-Ex. mean in line 101?
4. For the claim in line 168, is there any empirical support or reference? Also, what does the superscript star mean?
5. Does one need to create new domain templates when using the method in a new domain? How about mixed-domain applications?
6. Llama 3 isn't a reasoning model in Section 5.1.
7. What is the instruction given to the human evaluators?
8. Please add the dataset name in Figure 3's caption.
9. In Table 3, why CoT+RAG incurs +26s cost, while only RAG and only CoT both introduce 0s cost?
10. In Appx. A3, what is the data source for the data construction process?
11. The WTransition in line 161 might be a typo. Also, please check the grammar between lines 162-168. The format between lines 238-246 is also improvable.

---

### Official Review · Reviewer_urFP · 2025-11-01

**Soundness:** 2
**Presentation:** 3
**Contribution:** 3
**Rating:** 4
**Confidence:** 4

**Summary:**

This paper introduces LogicalChain + Test-Time-Training RAG (TTT-RAG), a framework that explicitly integrates structured logical chains relevant to the questions’ intent and perform test-time training to align model generation with retrieved reasoning. This paper also build large corpus of chains and train a contrastive retriever to fetch task-relevant inference paths. The proposed LogicalChain + TTT-RAG framework improves both medical and general multi-hop question answering performances.

**Strengths:**

- This paper introduces LogicalChain + TTT-RAG, which shows strong empirical gains on medical (MedQA, MedMCQA) and math domains, outperforming strong baselines such as MedRAG and rStar.
- This paper introduces a large corpus (2M+ pairs) of document and logical chain pairs from existing resources. Comparing to previous work that just focus on test-time retrieval augmentation for reasoning, this paper also train a retriever model tailored for this method.
- This paper trains a dual-stage retriever that retrieves and rerank logical chain instead of using off-the-shelf retriever models.
- Expert-annotated examples and manual spot-checks are used to validate the quality of retriever.
- Efficiency analysis are included in the paper.

**Weaknesses:**

- More details of the experiments are needed: What corpus do you use to evaluate MedRAG and other baselines? Are you using the provided corpus by them (namely wikipedia, pubmed, textbooks and statpearls)? Without ablations showing: (1) baseline methods using the same 2M+ corpus, or (2) TTT-RAG using baseline corpora, we cannot determine if the gains come from superior method design or simply having more/better training data. This is critical for establishing the method's actual contribution.
- Section 6.4 evaluates on math domains but: (1) never states whether math-specific logical chains were constructed or if medical chains were used; (2) provides no discussion of domain transfer or what 'robustness' means in this context; (3) shows weaker gains (+2.8 points on MATH) compared to medical domain (+16.3 on MedQA), suggesting potential overfitting. Please state clearly whether this is zero-shot domain transfer, or there is a separate math corpus.

**Questions:**

- Please fix the citation style (e.g., line 35-36; line 39-40). Use \citet and \citep accordingly.
- The citation for MedRAG is missing. There are actually multiple MedRAG online. If it is Xiong et al. (2024), it might be better to include the citation in line 149, where MedRAG first appears in the full text.
- In Table 9, do the textbooks and wikipedia corpora contain domains beyond medicine? What does “both” mean in this context?
- What fraction is synthetic vs. human-authored chains?

---

### Official Review · Reviewer_rTMX · 2025-11-04

**Soundness:** 2
**Presentation:** 1
**Contribution:** 2
**Rating:** 2
**Confidence:** 3

**Summary:**

This paper proposes TTT–RAG, an enhanced RAG method where authors 'test-time-train' a model to retrieve a set of query-relevant reasoning chains, producing a 'provisional' step-indexed chain. They then retrieve knowledge and verify at each transition before generating a final solution. They test with both simple CoT and RAG-augmented baselines, finding their method outperforms similar-sized models in both the medical domain and general purpose multi-hop reasoning tasks.

**Strengths:**

The step-wise idea of synthesizing candidate reasoning chains, and verifying reasoning iteratively (retrieve-and-verify at each transition) is interesting and seems especially relevant to domains such as medical, where verifiable logic is critical.

**Weaknesses:**

- The novelty of the test-time-training mechanism is unclear, and it’s not obvious why weight updates are needed if more conventional inference-time solutions (such as simple ICL) could achieve the same purpose. Extensive further analysis is required to better position and compare this this with other more conventional approaches.
- Missing some important analysis that would shed light into the behavior of their step-level retrieval verification mechanism. For example, how often and at what stages (ie, transitions) are failures most commonly observed?
- The feels hastily written and very difficult to follow, with many typos, grammar and presentation errors throughout.

Overall, the paper needs significant rewriting and clearer exposition.

**Questions:**

1. Section 6.1 / Table 1: how are the human ratings calculated? Is it a head-to-head / win rate versus your method? If not, can you provide the performance of your method?
2. Table 1: why report ROUGE instead of accuracy on MedQA?
3. How is step-level entailment (“atomic coverage”) calculated? What metric?
4. Section 6.4: are the ablations (small-batch size, parallel multi-query) performed only on math data? Why not on the initial domains? What are the range and increments you tested? You describe 1/2/4/8, yet the figure indicates 20–100 for retention percentage. Is there even a figure for the batch size ablation? The second figure is also missing a y-axis label.
4a.  In what scenarios do you believe it is reasonable to share gradient updates across queries (multi-query setting)? There seem to be many uncertainties and potential for malicious or adversarial attacks.


Presentation / Typos / Grammar Issues:

- l161: “Wtransition”
- l165: “Define a” — grammar
- Missing parentheses around the citations in the introduction.
- Figure 2: bottom-right of the diagram is particularly confusing.
- Figure 1: “Ours logical chain” — missing a colon. Additonally, please use consistent symbols/emoji and label these (e.g., indicate what is an LLM response vs a meta-comment vs an emoji just to highlight the behavior).
- Section title: “5 Experiment” → “5 Experiments?”
- Figure 3 caption: references a top and bottom? (seems like some your caption text, figures, and section text are misligned)
- Section 6.3: capitalize “Table”.
- Table 3: color-code the checks and x’s.

---

### Meta-Review · Area_Chair_wt2W · 2026-01-04

**Summary:**

Reviewers expressed concerns over the unclear novelty of test-time training compared to alternatives like ICL or SFT, unfair baselines and missing related work (e.g., R3-RAG), high inference costs, poor presentation with typos and confusing figures, and lack of detailed analyses on corpus quality, ablations, and failure modes, informing a suggested decision to reject the paper.

**Reviewer Concerns:**

No rebuttal is present in the provided document, so none of the concerns—such as novelty, baseline fairness, computational overhead, presentation errors, and missing analyses—have been addressed, leaving all outstanding.

**Reviewer Scores:**

Without a rebuttal or additional discussion, reviewers would likely retain their original scores, with 72Si and urFP at 4 (marginally below the acceptance threshold), Wg6F and rTMX at 2.

---

### Decision · Program_Chairs · 2026-01-26

Reject